# Burden, predictors, and outcome of unconsciousness among under-five children hospitalized for community-acquired pneumonia: A retrospective study from a developing country

Gazi Md. Salahuddin Mamun[1], Monira Sarmin[1], Abu Sadat Mohammad Sayeem Bin Shahid[1], Sharika Nuzhat[1], Lubaba Shahrin[1], Farzana Afroze[1], Haimanti Saha[1], Shamsun Nahar Shaima[1], Mst. Shahin Sultana[2], Tahmeed Ahmed[1], Mohammod Jobayer Chisti[1] *

1 Nutrition and Clinical Services Division, International Centre for Diarrhoeal Disease Research, Bangladesh (icddr,b), Dhaka, Bangladesh, 2 National Institute of Population Research and Training (NIPORT), Ministry of Health and Family Welfare, Dhaka, Bangladesh

* chisti@icddrb.org

**Data Availability Statement:** On the basis of recommendation of the Institutional Review Board,

## Abstract

Despite the reduction of death from pneumonia over recent years, pneumonia has still been the leading infectious cause of death in under-five children for the last several decades. Unconsciousness is a critical condition in any child resulting from any illness. Once it occurs during a pneumonia episode, the outcome is perceived to be fatal. However, data on children under five with pneumonia having unconsciousness are scarce. We've retrospectively analyzed the data of under-five children admitted at the in-patient ward of Dhaka Hospital of icddr,b during 1 January 2014 and 31 December 2017 with World Health Organization classified pneumonia or severe pneumonia. Children presented with or without unconsciousness were considered as cases and controls respectively. Among a total of 3,876 children fulfilling the inclusion criteria, 325 and 3,551 were the cases and the controls respectively. A multivariable logistic regression analysis revealed older children (8 months vs. 7.9 months) (adjusted odds ratio, aOR 1.02, 95% CI: 1.004–1.04, p = 0.015), hypoxemia (aOR 3.22, 95% CI: 2.39–4.34, p<0.001), severe sepsis (aOR 4.46, 95% CI: 3.28–6.06, p<0.001), convulsion (aOR 8.90, 95% CI: 6.72–11.79, p<0.001), and dehydration (aOR 2.08, 95% CI: 1.56–2.76, p<0.001) were found to be independently associated with the cases. The cases more often had a fatal outcome than the controls (23% vs. 3%, OR 9.56, 95% CI: 6.95–13.19, p<0.001). If the simple predicting factors of unconsciousness in children under five hospitalized for pneumonia with different severity can be initially identified and adequately treated with prompt response, pneumonia-related deaths can be reduced more effectively, especially in resource-limited settings.

the Research Administration of icddr,b has imposed a restriction on disclosing any personal information of hospitalized patients as this data contain sensitive patient information. However, data generated from icddr,b's electronic patient database can be provided to interested researchers for secondary data analyses upon approval of a Data Licensing Application & Agreement by the icddr,b Data Centre Committee. The data request may be sent to Ms. Armana Ahmed (aahmed@icddrb.org), Head, Research Administration.

**Funding:** The authors received no specific funding for this work.

**Competing interests:** The authors have declared that no competing interests exist.

## Introduction

Acute respiratory infections are one the most common under-five childhood illnesses that cause frequent visits to healthcare centers in low and middle-income countries (LMICs) [1]. Among these, pneumonia, an infection of lung parenchyma [2], is also a common reason for hospital admission in children in LMICs [1]. For the last several decades, pneumonia has remained the single leading infectious cause of death in under-five children [1,3,4]. Though death from pneumonia has been reduced due to effective interventions over recent years, still, pneumonia is responsible for 15% of an estimated 5.4 million global deaths in children under five years of age [3]. The greater burden of disease is concentrated in low-income countries, where resources for medications and hospital-based management are poor [5]. World Health Organization (WHO) includes a reduced level of consciousness as one of the danger signs of severe pneumonia aiming for early diagnosis and hospitalization of sick children so that pneumonia-related deaths can be reduced [6,7]. A study from a pediatric intensive care unit mentioned that those who had a new-onset seizure, among them 17% had complicated pneumonia [8]. Many of the children under five years of age also presented with unconsciousness during admission. However, the data on predictors behind this scenario and their outcomes are yet to be explored. A recent study among 2–59 months old has shown that mortality was significantly higher among the unconscious or decreased consciousness children hospitalized for pneumonia in 20 low- and middle-income countries including Bangladesh where 17.4% of children were abnormally sleepy [9,10]. But there is no known available data from Bangladesh regarding the burden and risk factors of unconsciousness among pneumonia children.

This analysis is aimed to find out the predictors of unconsciousness associated with WHO-classified pneumonia with different severity among under-five children. Our further aim was also to describe their seasonal variation in admission and outcomes at discharge from the hospital. By finding out potential predictors of unconsciousness in children under five with pneumonia having different severity, we may be able to have early diagnosis and prompt aggressive treatment that may help to reduce morbidity and mortality. Eventually, to attain the Global Action Plan for Pneumonia and Diarrhea, our findings from this study may help by eliminating pneumonia-related morbidity and mortality by the target year, 2025 [3].

## Methods and materials

### Study design

This is a retrospective chart analysis conducted between 1st January 2014 and 31st December 2017. We enrolled 0–59 months children with WHO-classified pneumonia with different severity [6,7] admitted to the ICU and ARI ward. Diagnosis of pneumonia was confirmed following WHO-defined radiologic pneumonia [11]. The study children who presented with unconsciousness, were included in this retrospective analysis and leveled as cases and those presented without unconsciousness were considered as controls in this study. Those without the data of consciousness level at admission were excluded. A comparison of socio-demographic and clinical characteristics on admission and outcomes on discharge was made between the cases and the controls.

### Study site and population

This study was conducted in the Dhaka Hospital of the International Centre for Diarrheal Disease Research, Bangladesh (icddr,b). This largest diarrheal hospital provides care and treatment for around 160,000 patients a year [12].

Since 2009, this hospital became paperless and uses an electronic patient medical record system to keep all clinical and laboratory data. Study patients were treated in the Intensive care Unit (ICU) and Acute Respiratory Infection (ARI) unit of the inpatient ward of the hospital. Ten-bedded Intensive Care Unit is dedicated to aiding medical care for critically ill patients presenting with respiratory distress and/or cyanosis, hypothermia, sepsis, severe sepsis, septic shock, altered mentation, convulsion, severe pneumonia, or respiratory failure along with diarrhea. This ward is enriched with necessary facilities for critical care management including pulse oximeters, cardiac monitors, bedside Glucometer, basic life support instruments, cardiac defibrillators, and noninvasive ventilation including locally made bubble continuous positive airway pressure (bCPAP) oxygen therapy [13], and mechanical ventilators, etc. ARI unit is situated within the Longer Stay Unit. It comprises 30 beds and is dedicated to patients admitted with respiratory infections including pneumonia. ARI unit is also well-equipped with a pulse oximeter, bedside glucometer, portable oxygen cylinders, oxygen concentrator, etc. A detailed description of the Dhaka Hospital of icddr,b has been written elsewhere [13,14].

## Patient management

All under-five children with pneumonia or severe pneumonia were given standard treatment, following hospital guideline [12] that was based on WHO guideline for pneumonia management [6]. Children who had severe respiratory distress or any danger sign received at least 4 hourly monitoring of clinical signs by physicians and nurses at the ICU. They received parenteral antibiotics and nasogastric feeding as appropriate. Locally made low-cost bubble CPAP device is part of standard care of this hospital and was received by the children who had severe pneumonia with hypoxemia or grunting or central cyanosis in absence of congenital heart disease. A randomized controlled trial in treating childhood severe pneumonia with hypoxemia by Chisti et al has shown the beneficial role (reduction of mortality and treatment failure) of this bubble CPAP device compared with WHO standard low-flow oxygen therapy [12]. Mechanical ventilation was considered for those who failed with bubble CPAP. Those who had pneumonia without any general danger signs were treated at the ARI unit and received at least 8 hourly follow-ups of clinical signs by nurses and physicians. In all of our study children having unconsciousness were treated as severe pneumonia and they were further evaluated for other etiology of unconsciousness and treated accordingly. Children under five presented with severe malnutrition [15], and diarrhea-related complications such as some to severe dehydration, and electrolyte imbalance [13] were treated according to protocolized management guidelines which are consistent with the WHO [6]. In addition to these, intravenous fluid resuscitation with isotonic fluid (either Hartmann's solution or Normal saline) was given for children with severe sepsis. Inotropes and vasopressors were given in case of septic shock [16].

## Data collection

A semi-structured case report form (CRF) was developed and finalized for the acquisition of study-relevant data from an electronic database. We recorded pneumonia-related events with and without unconsciousness at admission and their outcome during the hospital stay. We also documented the demographic information (age, sex), immunization status, clinical features, and laboratory findings of these under-five children. Clinical features included types of diarrhea, nutritional status, and dehydration, congenital heart disease (such as septal defects, tetralogy of Fallot, etc.), other congenital anomalies (such as cleft lip or palate, trisomy 21), hypoxemia (SpO2 <90% in room air), and severe sepsis. Laboratory investigations included hemoglobin level, total white blood cells (WBC) count, and blood culture and sensitivity report

those were done during hospitalization. The variables related to the outcome of all participants such as duration of hospital stay, respiratory failure, and death were also collected.

## Working definitions

**Unconsciousness.** Unconsciousness was defined in this study as having GCS 3 or children not responding to deep painful stimuli according to the AVPU (alert, verbal, pain, unresponsive) criteria at admission.

**Pneumonia and severe pneumonia.** These were defined according to WHO classification. In presence of cough or difficulty in breathing: if a child had age-specific fast breathing or chest wall indrawing, then they were classified as pneumonia; if a child had any one of these, then they were classified as severe pneumonia: (i) Oxygen saturation <90% or central cyanosis, (ii) Grunting respiration (iii) WHO defined general danger sign (inability to breastfeed or drink, lethargy or reduced level of consciousness, convulsions) [7].

**Diarrhea and dehydration.** Diarrhea was defined as the passage of three or more abnormally loose stools per 24 hours [7]. It has three clinical types such as acute watery diarrhea (the most common type), invasive diarrhea (defined as the passage of visible bloody mucoid stool one or more times within 24 hours), and persistent diarrhea (passage of ≥3 loose stools for consecutive ≥14 days from an acute onset) [7,17]. Dehydration was assessed by the Dhaka method almost consistent with WHO methods and classified as no dehydration, some dehydration, and severe dehydration [18,19].

**Severe sepsis.** Severe sepsis was defined according to surviving sepsis guideline with mild modification for diarrheal patients [16,20].

**Severe acute malnutrition.** For nutritional status, we evaluated the weight-for-length/ height Z-score and the presence or absence of nutritional edema. If the weight-for-length Z-score was <-3 SD of WHO median or had edema of both feet in our study children, they were considered to have severe acute malnutrition (SAM) [7].

**Data analysis.** Data extracted from the electronic database of icddr,b, was entered into SPSS for Windows (version 20.0; SPSS Inc, Chicago). Data cleaning and corrections were done in cases of inconsistencies or errors in the entered data. Such as incomplete or short keywords or diagnoses were corrected or rephrased manually by expert clinicians after checking all the records from the electronic database for statistical analysis. All statistical analyses were done using STATA SE 15.0 (Texas, USA) software. Clinical, socio-demographic, laboratory, and other relevant data were summarized using descriptive statistics. Regarding continuous variables, means with standard deviations (SD) were used in case of normally distributed data, and medians with interquartile ranges (IQRs) were used in case of skewed data. Bivariate logistic regression analysis was done to find out the associated predictors of unconsciousness. Multivariable logistic regression analysis was done to find out the independently predicting factors of unconsciousness. In the regression model, unconsciousness was the dependent variable whereas significantly associated factors with unconsciousness (having p-value <0.1 from the bivariate analysis of those clinical features presented on admission) were the independent variables. We have also put age and sex in the multivariable regression model as the confounding variables due to their biological plausibility with unconsciousness. The strength of association was estimated by evaluating the odds ratio (OR). In this study, a 95% confidence interval and a p-value <0.05 were considered statistically significant values.

**Ethical considerations.** Data were extracted from electronic medical records of patients hospitalized in the intensive care unit and acute respiratory infection unit. The patients' information was anonymized and de-identified before analysis and did not include any direct interview with the caregivers. Still, a waiver of the ethical approval of hospital data disclosure for

this analysis was obtained from the Institutional Review Board (IRB) of the International Centre for Diarrhoeal Disease Research, Bangladesh (icddr,b) that comprises of Research Review Committee (RRC) and Ethical Review Committee (ERC).

## Results

During the study period, a total of 4,007 under-five children were admitted due to WHO-classified pneumonia with different severity. Among them, 131 (3.27%) participants didn't have the data on consciousness level during admission. Among the remaining 3,876 participants, 325 (8.38%) were unconscious and 3,551 (91.62%) were conscious (Fig 1).

Compared with the children with consciousness, those who were unconscious during admission, they were more likely to be female (OR 1.41, 95% CI: 1.12–1.78, p = 0.004), often presented with hypoxemia (OR 8.6, 95% CI: 6.65–11.14, p<0.001), severe sepsis (OR 10.31, 95% CI: 7.96–13.34, p<0.001), acute watery diarrhea (OR 2.23, 95% CI: 1.52–3.26, p<0.001), invasive diarrhea (OR 2.23, 95% CI: 1.17–4.24, p = 0.015), dehydration (OR 2.66, 95% CI: 2.10–3.36, p<0.001) and history of convulsion (OR 15.97, 95% CI: 12.42–20.53, p<0.001) on admission (Table 1). Children having congenital anomalies were less likely to present with unconsciousness during admission (OR 0.32, 95% CI: 0.14–0.74, p = 0.007). Other variables such as severe acute malnutrition, immunization status, presence of congenital heart disease, and persistent diarrhea were found to be comparable between the groups (Table 1).

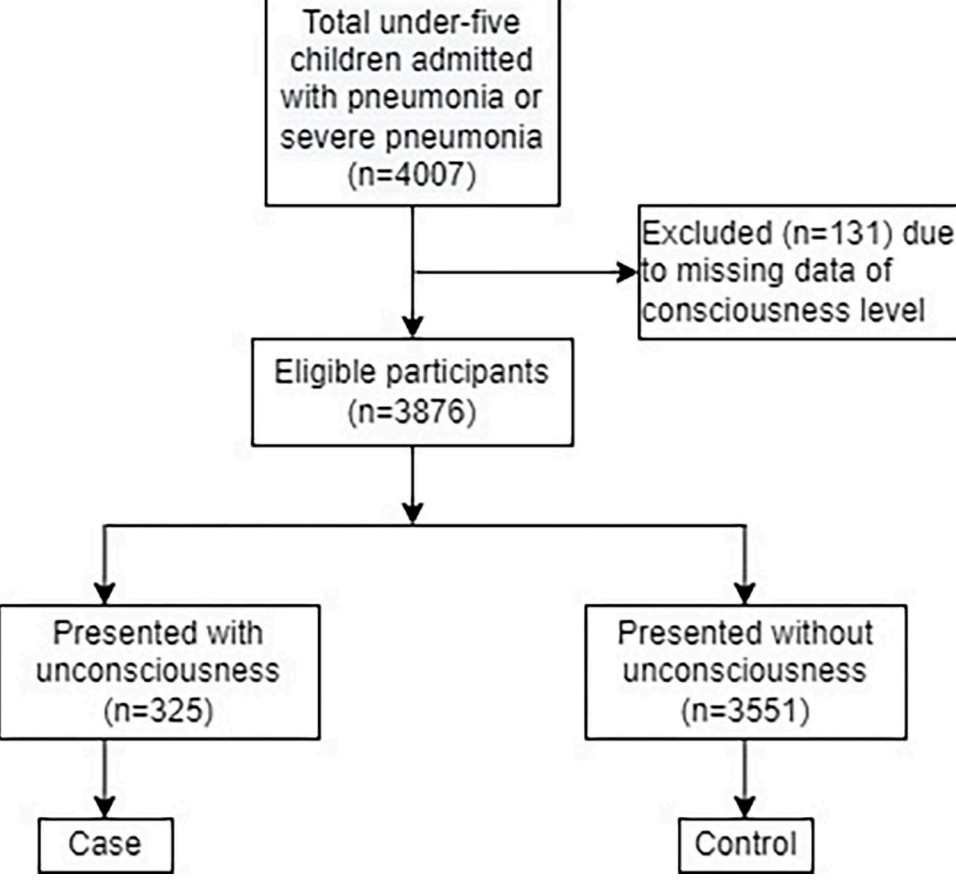

**Fig 1. Study diagrams displaying the selection of participants as cases and controls.**

**Table 1. Admission characteristics of under-five children hospitalized for WHO-classified pneumonia with different severity and with or without unconsciousness during admission.**

| Characteristics | Children with unconsciousness (n = 325) | Children without unconsciousness (n = 3551) | OR (95% CI) (unadjusted) | P value |
|---|---|---|---|---|
| Age (months) (median, IQR) | 8.0 (4.59, 11.59) | 7.92 (4.89, 12) | 1.00 (0.98–1.01) | 0.945 |
| Sex (female) | 135 (41.5) | 1190 (33.5) | 1.41 (1.12–1.78) | 0.004 |
| Hypoxemia | 239 (73.5) | 867 (24.4) | 8.60 (6.65–11.14) | <0.001 |
| SAM[1] | 131 (40.3) | 1518 (42.8) | 0.90 (0.72–1.14) | 0.394 |
| Immunization | 214/269 (79.6) | 2222/2710 (82.0) | 0.85 (0.63–1.17) | 0.324 |
| CHD[2] | 23 (7.1) | 234 (6.6) | 1.08 (0.69–1.68) | 0.735 |
| Severe sepsis | 135 (41.54) | 229 (6.5) | 10.31 (7.96–13.34) | <0.001 |
| Type of diarrhea | | | | |
| No diarrhea | 31 (9.5) | 654 (18.4) | Reference | |
| AWD[3] | 274 (84.3) | 2593 (73.0) | 2.23 (1.52–3.26) | <0.001 |
| ID[4] | 15 (4.6) | 142 (4.0) | 2.23 (1.17–4.24) | 0.015 |
| PD[5] | 5 (1.5) | 162 (4.6) | 0.65 (0.25–1.70) | 0.381 |
| Dehydration | 133 (40.9) | 734 (20.7) | 2.66 (2.10–3.36) | <0.001 |
| Convulsion | 191 (58.8) | 291 (8.2) | 15.97 (12.42–20.53) | <0.001 |
| Congenital anomaly | 6 (1.9) | 195 (5.5) | 0.32 (0.14–0.74) | 0.007 |
| Hemoglobin (gm/dL, mean±SD) | 10.3 ± 2.2 (n = 301) | 10.7 ± 2.9 (n = 2,985) | 0.91 (0.86–0.96) | 0.001 |
| WBC count (cc/HPF, median, IQR) | 16,190 (11,910, 22,080) (n = 301) | 14,210 (10,220, 19,000) (n = 2,985) | 1.000023 (1.000013–1.000034) | <0.001 |
| Blood CS positive | 42/254 (16.5) | 224/1526 (14.7) | 1.14 (0.80–1.63) | 0.533 |

[1]SAM- Severe acute malnutrition

[2]CHD- Congenital heart disease

[3]AWD- Acute watery diarrhea

[4]ID- Invasive diarrhea

[5]PD- Persistent diarrhea.

Complete blood count including hemoglobin level and WBC count was done on 3,286 participants (case 92.6% vs control 84.1%) and blood CS was done on 1,780 participants (case 78.2% vs control 43.0%). For those who presented as unconscious, their hemoglobin level was found significantly lower (10.3 ± 2.2 vs. 10.7 ± 2.9; p = 0.001) and WBC count was significantly higher [16,190 (11,910, 22,080) vs. 14,210 (10,220, 19,000); p<0.001] compared to those who presented without unconsciousness (Table 1).

Regarding the outcomes, children who presented with unconsciousness had higher odds of developing respiratory failure (OR 17.04, 95% CI: 12.42–23.38, p<0.001) and death (OR 9.56, 95% CI: 6.95–13.19, p<0.001) compared to the conscious children at admission (Table 2). In this table, the duration of hospital stay as an outcome was also included, but no significant difference was found between cases and controls.

**Table 2. Outcome of under-five children hospitalized for WHO classified pneumonia with different severity and presented with or without unconsciousness during admission.**

| Characteristics | Children with unconsciousness (n = 325) | Children without unconsciousness (n = 3551) | OR (95% CI) (unadjusted) | P value |
|---|---|---|---|---|
| Hospital stay in days (median, IQR) | 6 (3, 8) | 6 (4, 8) | 0.99 (0.98–1.01) | 0.582 |
| Respiratory failure | 99 (30.5) | 89 (2.5) | 17.04 (12.42–23.38) | <0.001 |
| Death | 75 (23.08) | 108 (3.04) | 9.56 (6.95–13.19) | <0.001 |

**Table 3. Results of multivariable logistic regression analysis to explore the independent predictors of unconsciousness in under-five children hospitalized for WHO-classified pneumonia with different severity.**

| Characteristics | aOR | 95% CI | P value |
|---|---|---|---|
| Age (months) | 1.02 | 1.004–1.04 | 0.015 |
| Sex (female) | 1.21 | 0.92–1.59 | 0.171 |
| Hypoxemia | 3.22 | 2.39–4.34 | <0.001 |
| Severe sepsis | 4.46 | 3.28–6.06 | <0.001 |
| Convulsion | 8.90 | 6.72–11.79 | <0.001 |
| Dehydration | 2.08 | 1.56–2.76 | <0.001 |
| Congenital anomaly | 0.27 | 0.11–0.66 | 0.004 |

After adjusting the potential confounders, in multivariable logistic regression analysis, older children (8 months vs. 7.9 months) (aOR 1.02, 95% CI: 1.004–1.04, p = 0.015), hypoxemia (aOR 3.22, 95% CI: 2.39–4.34, p<0.001), severe sepsis (aOR 4.46, 95% CI: 3.28–6.06, p<0.001), convulsion (aOR 8.90, 95% CI: 6.72–11.79, p<0.001), and dehydration (aOR 2.08, 95% CI: 1.56–2.76, p<0.001) were found as significant independent predictors of unconsciousness in under-five children hospitalized with pneumonia or severe pneumonia. On the other hand, those who had congenital anomaly (aOR 0.27, 95% CI: 0.11–0.66, p = 0.004), had significantly less chance of being presented as unconscious (Table 3). Although older age was not significant in bivariate analysis, it was found as an independent predictor of unconsciousness after adjusting with other significant variables.

Regarding the seasonal variations of unconsciousness among children admitted with WHO-classified pneumonia with different severity, the percentage of unconsciousness was more during the monsoon, pre-winter, and winter seasons (July to January). In 2014, there was an unusual peak during summer (mid-April to mid-June) (Fig 2).

## Discussions

To our knowledge, this is the first study that evaluated the predicting factors and outcome of unconscious children under five years of age hospitalized for WHO-classified pneumonia with different severity. The main observation of the study is the higher rate of deaths, hypoxemia, severe sepsis, convulsion, and dehydration among the study children having unconsciousness compared to those who were conscious. This study also identified the burden of unconsciousness in our study children which was 8.38%.

Though the prevalence of unconsciousness in pneumonia looks relatively low considering other variables, still unconsciousness has a highly significant effect on morbidity and mortality. We've found that children who presented with unconsciousness had higher odds of developing both respiratory failure and death compared to the conscious children at admission.

Since 1990, pneumonia-related mortality has been reduced to 67%. But even after treatment according to the WHO recommended guidelines such as proper antibiotics, oxygen therapy, and other adequate care, mortality in under-five children with pneumonia, especially in developing countries remains high [21–23]. Those children who are at higher risk of death despite receiving internationally recommended interventions may further improve the poor outcomes due to pneumonia. These under-five children hospitalized for WHO-classified pneumonia with different severity having unconsciousness than their counterpart is found to have a higher risk of death. A study conducted by Tiewsoh et al., among 200 under-five children having severe pneumonia during 2004–06 in India, has shown that altered mental status that includes unconsciousness was associated with increased mortality [24]. Jakhar et al. also conducted a

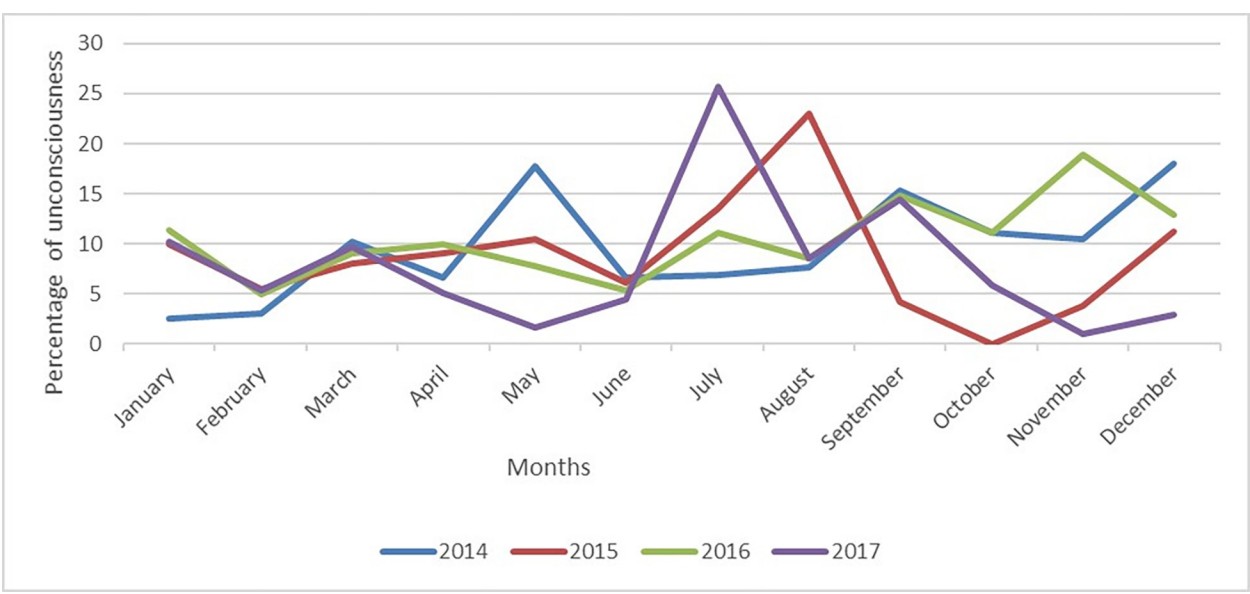

**Fig 2. Seasonal variations of unconsciousness among children under five years of age admitted with WHO-classified pneumonia with different severity during 2014–2017.**

study in India among under-five children with pneumonia during 2012–2013 and found that children with lethargy or unconsciousness succumbed to treatment failure [25].

We've also found that hypoxemia and convulsion are independently associated with unconsciousness among under-five children. Studies have shown that neuronal cell death occurs due to cerebral hypoxia-ischemia [26,27], which may lead to convulsion or even unconsciousness. Studies have shown that hypoxemia and ischemia can cause seizures in young children [28]. Hypoxemia is also associated with higher mortality in acute lower respiratory infections in children from LMICs [29]. Another study has shown that hypoxemia among under-five diarrheal children with or without pneumonia had a higher risk of fatal outcomes in an urban hospital in Bangladesh [30]. But none of these studies specifically showed any association with unconsciousness.

We've also found a significant association of having unconsciousness with severe sepsis compared to those who didn't have unconsciousness. A study by Chisti et al. has shown that 62.2% of infants suffering from diarrhea and systemic inflammatory response syndrome, presented with altered mental status [31]. Another study has shown that drowsiness is also significantly higher among severe sepsis under-five children who presented with pneumonia and severe acute malnutrition [20].

Dehydration is also found as a risk factor for unconsciousness in this study. According to WHO guidelines, a severely dehydrated child may present with unconsciousness [7].

In this study, a higher risk of unconsciousness among relatively older children is found. This may be due to more awareness and early health-seeking behavior of parents in case of younger children. We do not have any ready explanation for this observation. However, Chisti et al have shown no age difference in mortality among children with SAM presenting with cough or respiratory difficulty and radiological pneumonia [22]. Chakraborty et al also showed no difference in pneumonia mortality related to age among under-five children from West Bengal [32]. Further study is needed to find out the cause behind this observation.

Children having congenital anomalies were relatively conscious at admission. Earlier motivation and health awareness of parents during the diagnosis of the congenital anomaly might

compel them to seek health care early for their vulnerable children. Several studies have shown that though not always adequate [33], parents of children having congenital anomalies used to receive various types of health awareness [34,35], and parents are also interested to know in detail about their children's health problems [36]. Another reason behind this may be that pneumonia in children with congenital anomalies might not be due to the same etiology compared to those without congenital anomalies. But the specific reason is still to be found out.

The seasonality of unconsciousness in pneumonia during the monsoon and winter seasons is unclear. No relevant study related to this is found. This might be due to the fact that the organisms that were responsible for unconsciousness in pneumonia among under-five children may be more prevalent during this period. But we don't have any data related to this. Though the seasonality of pneumonia is known, but further study may be needed to determine the causes of the seasonality of unconsciousness in children with pneumonia.

## Limitations of the study

It was a single-center study and retrospective in nature. There were limited facilities for the elaboration of laboratory investigations to ascertain the cause of unconsciousness other than severe pneumonia. The lack of data on the progression of pneumonia to severe pneumonia is another limitation of this study.

## Strength of the study

The main strength of this study is the large sample size having four years of childhood pneumonia data where the diagnosis of pneumonia/severe pneumonia following WHO classification was confirmed by pediatricians (LS, FA, and HS) and pediatric respiratory physician (MJC).

## Conclusions

This study has found a significant association of death with unconsciousness among under-five pneumonia children than those admitted with consciousness. We have also identified hypoxemia, severe sepsis, convulsion, and dehydration as simply preventable predictors of unconsciousness in our study children. If these preventable factors can be spotted earlier with prompt and proper medical intervention with health education or counseling about these high-risk factors, then death due to pneumonia may be substantially reduced. However, prospective longitudinal study is imperative to understand in-depth risk factors of unconsciousness, that may help to prevent this fatal ramification of pneumonia in children under five years of age.

## Supporting information

**S1 Text. STROBE checklist for an observational study.**
(DOCX)

## Acknowledgments

We gratefully acknowledge our core donors for their support and commitment to icddr,b's research efforts. Current donors providing unrestricted support include the Governments of Bangladesh, Canada, Sweden, and the UK. We would also like to express our sincere thanks to all clinical fellows, nurses, members of the feeding team, and cleaners of the hospital for their invaluable support and contribution to patient care.

## Author Contributions

**Conceptualization:** Gazi Md. Salahuddin Mamun, Mst. Shahin Sultana, Mohammod Jobayer Chisti.

**Data curation:** Gazi Md. Salahuddin Mamun, Shamsun Nahar Shaima, Mohammod Jobayer Chisti.

**Formal analysis:** Gazi Md. Salahuddin Mamun, Monira Sarmin.

**Methodology:** Gazi Md. Salahuddin Mamun, Mohammod Jobayer Chisti.

**Supervision:** Mohammod Jobayer Chisti.

**Validation:** Gazi Md. Salahuddin Mamun.

**Visualization:** Gazi Md. Salahuddin Mamun, Monira Sarmin, Tahmeed Ahmed, Mohammod Jobayer Chisti.

**Writing – original draft:** Gazi Md. Salahuddin Mamun.

**Writing – review & editing:** Gazi Md. Salahuddin Mamun, Monira Sarmin, Abu Sadat Mohammad Sayeem Bin Shahid, Sharika Nuzhat, Lubaba Shahrin, Farzana Afroze, Haimanti Saha, Shamsun Nahar Shaima, Mst. Shahin Sultana, Tahmeed Ahmed, Mohammod Jobayer Chisti.

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
