## [Decision Letter · Decision Letter 0]

20 Apr 2023

PONE-D-23-04422Burden, predicting factors and outcome of unconsciousness among under five children hospitalized for community-acquired pneumonia

Dear Dr. Chisti,

Thank you for submitting your manuscript to PLOS ONE. After careful consideration, we feel that it has merit but does not fully meet PLOS ONE’s publication criteria as it currently stands. Therefore, we invite you to submit a revised version of the manuscript that addresses the points raised during the review process.

We look forward to receiving your revised manuscript.

Kind regards,

Sanjoy Kumer Dey, M.D

Academic Editor

PLOS ONE

Journal Requirements:

. We note that you have indicated that data from this study are available upon request. PLOS only allows data to be available upon request if there are legal or ethical restrictions on sharing data publicly. For more information on unacceptable data access restrictions, please see http://journals.plos.org/plosone/s/data-availability#loc-unacceptable-data-access-restrictions. 

Reviewers' comments:

Reviewer's Responses to Questions

**Comments to the Author**

1. Is the manuscript technically sound, and do the data support the conclusions?

Reviewer #1: Yes

Reviewer #2: Partly

2. Has the statistical analysis been performed appropriately and rigorously? 

Reviewer #1: No

Reviewer #2: Yes

3. Have the authors made all data underlying the findings in their manuscript fully available?

Reviewer #1: Yes

Reviewer #2: Yes

4. Is the manuscript presented in an intelligible fashion and written in standard English?

Reviewer #1: Yes

Reviewer #2: Yes

5. Review Comments to the Author

Reviewer #1: The authors primarily focused on the burden, predicting factors and outcome of unconsciousness among under five children hospitalized for community-acquired pneumonia, which I believe to be an important topic. The paper is interesting and generally well-written. I read the manuscript thoroughly, and my feedback as followed.

This is a retrospective case-control study, it is unclear that authors mention predicting factors instead of risk factors in title.

In line 48, it is unclear WHO. In line 55 to 60, authors find out predicting factors instead of risk factors in the retrospective case-control study. Author’s aim should identify the risk factors.

Authors miss the study site or country name in title and do not convey any situation of unconsciousness in their study areas in the introduction section. I found a typo mistake “ration” in line 77. This is a retrospective chart review from a historical record, it is not a match case control study. it is unclear the statement in line 77-78 “Though the case control ratio 1:10, …………………………………..not excluded the participants beyond 1:4 ratio.” I think that all eligible participants must be included in the study. Authors used chi-square test for examining the association between unconsciousness and pneumonia, but this test is not used to compare between two categorical variables (line # 171). In Data analysis section, authors unnecessarily mention chi-square test, Student’s t-test, and Mann-Whitney Test. Authors sometimes mention associated factors (line# 175) in the data analysis. Authors must select factors those were significant at level 10% for multivariate analysis. authors missed this point. In table 1, authors missed the odds ratio (OR) for age, Hemoglobin, and WBC count. Age, Hemoglobin, and WBC count are continuous but it is possible to calculate OR by using logistic regression. In line # 224, authors need to mention confounding variables. In Table 3, Multivariate regression analysis is used to adjust confounding variables in the relation between outcome and exposure. Authors used multiple logistic regression analysis to identify the risk factors of unconsciousness. All variables were adjusted for each other. It is unexpected to adjust hypoxemia, severe sepsis and so on for age and sex to estimate OR. It is good to adjust age and sex as confounding variables to estimate OR for hypoxemia or severe sepsis by using multivariate regression analysis. All results will be contradictory before fit best regression model. In conclusion, authors do not mention the preventable factors of unconsciousness.

Reviewer #2: Major comments

1. The authors mentioned unconsciousness among under-five children hospitalized for community-acquired pneumonia. Regarding unconsciousness, though the authors mentioned the percentage of pneumonia and severe pneumonia, however, the authors suggested stating how many of them developed severe pneumonia from pneumonia after admission (as it is a danger sign). It will provide a clear idea.

2. The inclusion and exclusion criteria for the study have to be clearly stated. For example, age 0-59 months or 2-59 months.

3. In L-166, the authors mentioned, data cleaning and corrections were done in cases of inconsistencies or errors in the entered data. Due to the retrospective nature of the data, how did they clean it up and fix the inconsistencies? It needs to explain clearly.

4. It's unclear why the authors included the ‘Hospital stay in days’ variable in Table 2. Better to provide an explanation.

5. In L-224-225, and L-284 the authors mentioned older children, but they did not mention the age group of the children before. Need to correct/revise it.

6. In Figure 2, the authors have shown only the months over the years. It is recommended to mention seasonality (monsoon, pre-winter, winter season) along with the month.

7. In L-290-292, the authors mentioned ‘Earlier motivation and health awareness of parents during the diagnosis of the congenital anomaly might compel them to seek health care early for their vulnerable children’. It would be better if they could give literature support/reference regarding this statement.

8. In L-298-300, the author could revise the sentences like this: The authors could write that the seasonality of pneumonia is known, but further study may be needed to determine the causes of the seasonality of unconsciousness in children with pneumonia.

9. In L-284-289, these sentences seem inconsistent with the study’s result. Please revise it.

10. In L-298, it is not about unconsciousness but also about pneumonia. Please rewrite the sentence.

11. Please add recent references (preferably the last 5-7 years) and follow the journal guidelines. The text citation would be [ ], not ( ). Please check the journal guidelines thoroughly.

12. The future recommendation is not strong. Please add a sentence about how these observations can be sustainable.

Minor comments

1. In L-41, in the first sentences, please add a citation.

2. In L-48, WHO needs to be abbreviated as it is used for the first time.

3. In L-84, please omit the word operated and replace it with conducted.

4. In L-86-88, the authors are suggested to omit this sentence.

5. In L-146, in oxygen words ‘O’ will be capitalized as the next words followed so. Please make it consistent.

6. In L-192-195, the authors are suggested to omit these sentences as this information was already mentioned in the Method section.

7. In Table 2, the authors need to mention whether the OR is unadjusted, as they mentioned in the rest of their tables. Please make it consistent.

8. The authors could add more points regarding strengths.

9. Please check the English grammar all over the manuscript.

6. PLOS authors have the option to publish the peer review history of their article (what does this mean?). If published, this will include your full peer review and any attached files.

Reviewer #1: No

Reviewer #2: **Yes: **Yasmin Jahan

---

## [Author Response · Author response to Decision Letter 0]

24 May 2023

Reviewer 1: All comments are addressed and mentioned in the "Response to reviewers" file.

Reviewer 2: All comments are addressed and mentioned in the "Response to reviewers" file.

Academic editor: Revised files are submitted according to the academic editor's suggestions.

---

## [Editor Report · Decision Letter 1]

29 May 2023

Burden, predictors, and outcome of unconsciousness among under-five children hospitalized for community-acquired pneumonia: A retrospective study from a developing country

PONE-D-23-04422R1

Dear Dr. Mohammod Jobayer Chisti

We’re pleased to inform you that your manuscript has been judged scientifically suitable for publication and will be formally accepted for publication once it meets all outstanding technical requirements.

Kind regards,

Sanjoy Kumer Dey, M.D

Academic Editor

PLOS ONE

---

## [Editor Report · Acceptance letter]

6 Jun 2023

PONE-D-23-04422R1 

Burden, predictors, and outcome of unconsciousness among under-five children hospitalized for community-acquired pneumonia: A retrospective study from a developing country 

Dear Dr. Chisti:

I'm pleased to inform you that your manuscript has been deemed suitable for publication in PLOS ONE. Congratulations! Your manuscript is now with our production department. 

Kind regards, 

on behalf of

Dr. Sanjoy Kumer Dey 

Academic Editor

PLOS ONE